# SALSA-V: SHORTCUT-AUGMENTED LONG-FORM SYNCHRONIZED AUDIO FROM VIDEOS

## ABSTRACT

We propose SALSA-V, a multimodal video-to-audio generation model capable of synthesizing highly synchronized, high-fidelity long-form audio from silent video content. Our approach introduces a masked diffusion objective, enabling audio-conditioned generation and the seamless synthesis of audio sequences of unconstrained length. Additionally, by integrating a shortcut loss into our training process, we achieve rapid generation of high-quality audio samples in as few as eight sampling steps, paving the way for near-real-time applications without requiring dedicated fine-tuning or retraining. We demonstrate that SALSA-V significantly outperforms existing state-of-the-art methods in both audiovisual alignment and synchronization with video content in quantitative evaluation and a human listening study. Furthermore, our use of random masking during training enables our model to match spectral characteristics of reference audio samples, broadening its applicability to professional audio synthesis tasks such as Foley generation and sound design.

## 1 INTRODUCTION

Video-to-audio (V2A) generation, sometimes referred to as "computational Foley", aims to produce realistic sounds for the visual events occurring in a silent video clip. Unlike background music or speech synthesis, Foley focuses on diegetic sounds, which are sounds implied by the current on-screen content (e.g., the sound of rain and thunder when a storm is shown, or a dog's bark echoing in a room). Achieving realism requires semantic (the model must recognize what is happening so it can select the right acoustic event) as well as temporal alignment (it must identify when that event occurs). Especially temporal alignment is crucial, as humans are sensitive to as few as tens of milliseconds of asynchrony (Keetels & Vroomen, 2005).

Early generative machine learning models for video-to-audio were trained from scratch on modestly-sized audio-visual corpora and struggled to cover the acoustic diversity of in-the-wild video. Recent work has addressed this issue by borrowing scale from adjacent modalities. One line of research adapts large text-to-audio diffusion models with lightweight video-conditioned control modules (Zhang et al., 2024), while another trains end-to-end multimodal generators using cross-modal attention, latent diffusion, or rectified-flow objectives (Cheng et al., 2025; Wang et al., 2025).

While recent approaches have resulted in significant improvements being made in this domain, existing models still face issues with precise on-screen synchronization and controllability of the generated sounds. Additionally, most existing works focus solely on generating audio for short, isolated snippets, often limited to a maximum length of 10 seconds due to either fundamental architectural restrictions or memory constraints. Attempting to utilize these approaches for longer sequences often significantly deteriorates performance, or can lead to stitching artifacts if shorter sequences are processed independently and later re-joined. In addition, users are offered limited control over the generated sounds, with most models opting for text conditioning for this purpose. Apart from limiting sound characteristics to those describable through text, this type of conditioning by itself does not natively support any editability of the produced audio. Furthermore, generation speed is usually treated as an afterthought by most approaches, requiring (in the case of diffusion models) many sampling steps to achieve high-quality results. This restricts the usability of these models to slow-feedback workflows and offline-only uses.

Our work aims to address these aspects, achieving the ability to handle audio conditioning and extended-length generations without large reductions in quality. Furthermore, our model enables near-real-time high-quality audio generations by using a dedicated training paradigm that enables few-step sampling natively (i.e., without special fine-tuning or re-training). We extensively evaluate our model against the existing state-of-the-art, including conducting a human preference study.

Our contributions can be summarized as follows:

- We introduce SALSA-V, a shortcut-augmented latent flow matching model for video-to-audio, enabling high-fidelity few-step sampling without distillation or additional fine-tuning.

- Support for audio conditioning and outpainting through a masked training objective, enabling the generation of samples based on reference audio sections and long-form generations using iterative extension.

- A contrastively-trained audio-visual alignment model using a backbone with large-scale pretraining, which yields high-resolution synchronization features, and results in state-of-the-art temporal synchronization and strong human MOS.

**Samples are available at** `https://anonymous.4open.science/w/salsav`.

## 2 RELATED WORK

### 2.1 VIDEO-TO-AUDIO GENERATION

Early efforts approached video-to-audio generation as a self-supervised prediction problem. Visually Indicated Sounds (Owens et al., 2016) synthesized impact noises from silent video using an RNN trained to regress sound features, demonstrating that visual cues can encode material and action properties. With the advent of more capable generative models, the focus has recently shifted to high-fidelity, temporally aligned synthesis. Diff-Foley (Luo et al., 2023) was the first work to focus on video-to-audio generation with diffusion models, demonstrating the feasibility of this approach. Frieren (Wang et al., 2025) replaced the standard diffusion training process with a rectified flow matching approach (Liu et al., 2022). By using model distillation (reflowing the original model), they can reduce the number of sampling steps necessary, but do so by re-training and distilling their model in dedicated fine-tuning phases. Complementarily, MMAudio (Cheng et al., 2025) introduced multimodal joint training on large text–audio corpora and a frame-level synchronization module, achieving state-of-the-art results, both in terms of general audio quality, as well as synchronization to onscreen events. An autoregressive approach was recently explored in V-AURA (Viertola et al., 2024), which natively enables long-form generation capabilities. However, V-AURA's approach is somewhat limited in that it does not allow for text conditioning and is restricted to a comparatively short context window of 2.56 seconds.

### 2.2 CROSS-MODAL AUDIOVISUAL LEARNING

Existing multimodal representation learning approaches focus mostly on learning similarities between visual features and audio in a "global manner", meaning that the temporal information inherent in both modalities is either ignored completely or treated coarsely (e.g. by subsampling videos at a low frame rate). These approaches include ImageBind (Girdhar et al., 2023), which learns a latent space covering various different modalities, and VATT (Akbari et al., 2021). More recently, foundational models, such as InternVideo2 (Wang et al., 2024), have also been used for this purpose.

These approaches are highly capable of learning broad semantic similarities, but are not concerned with high-resolution temporal alignment between individual video frames and audio. Another line of work does place their focus on this topic, including Synchformer (Iashin et al., 2024), which has a contrastive pretraining stage (*Segment-AVCLIP*) trained on short video-audio snippets. These embeddings are then used to predict a global synchronization score between a given audio and video segment. The embeddings learned by Synchformer's first stage have been utilized by previous V2A models to provide synchronization signals for downstream generative models (Cheng et al., 2025).

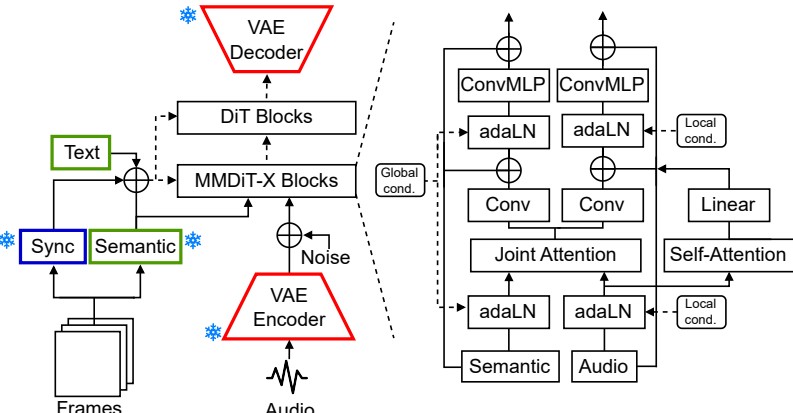

Figure 1: Architectural diagram of SALSA-V. We utilize a mixture of modified MMDiT-X blocks operating jointly over the combined sequence of audio and semantic features, as well as single-stream DiT blocks, which operate on the latent representation of a VAE (red). The overall global conditioning signal consists of the text embedding and pooled semantic features (green), which are extracted from a pretrained SigLIP 2 encoder, synchronization features (blue) obtained from a dedicated contrastive model, and the time-step embedding. The local conditioning tensor equals the global conditioning summed to the sequence-aligned synchronization features.

## 2.3 EFFICIENT GENERATIVE MODELS

Due to the high number of sampling steps required for high-quality generative results when using diffusion models (usually dozens in most domains), most generative audio models are not suited for real-time applications. Consequently, there has recently been some interest in improving this aspect by reducing the number of sampling steps necessary to achieve higher fidelity. In diffusion and flow-matching models, this is commonly done by employing additional fine-tuning stages, where the denoising trajectory of the model is straightened (Liu et al., 2022). This enables few-step generations with significantly lower errors by letting the model take larger straight-line steps (along the newly learned path) before having to recalculate the velocity field. While multiple of these approaches have been successfully employed in the image generation domain, the only video-to-audio generation model which employs such a strategy is Frieren (Wang et al., 2025), which uses a rectified flow training objective (Liu et al., 2022).

## 2.4 LONG-FORM AUDIO GENERATION

While standard V2A models are not in principle restricted to inputs of a few seconds, in practice, memory (growing with the square of the sequence length in transformer models) and data constraints lead to this restriction being adopted (often resulting in maximum model inputs of 10 seconds). This leads to models that only perform well on this limited set of durations, with many experiencing a sharp drop-off in performance when extended even modestly beyond that point. LoVA (Cheng et al., 2024) first placed a focus on long-form generation. The authors attempt to address this issue by using a standard diffusion transformer (Peebles & Xie, 2023), but training it on dedicated long-form data (up to 30 seconds). Although this leads to performance improvements over training solely with short, fixed-size snippets, it does not address the underlying issue of unrestricted memory growth, and the generated audio snippets are still limited by the maximum length observed during training.

## 3 METHODOLOGY

A high-level schematic of our model and training process can be seen in Figure 1. We use a latent flow matching approach, utilizing an audio encoder based on Stable-Audio VAE (Evans et al., 2024), which was fine-tuned to increase its frame rate to 43 Hz in order to achieve higher audio quality (see Appendix A.5). We denote this input latent sequence as $x_1$. Its shape is $(t_a, 64)$, where $t_a$ is

the encoded sequence length. $x_1$ then undergoes the flow-matching noise-injection process on all sequence items, except for a randomly sampled contiguous subsequence (see Section 3.2). $x_t$ is then processed through a diffusion transformer model (Peebles & Xie, 2023), which employs a similar architecture to recent multi-modal diffusion models (Cheng et al., 2025; Esser et al., 2024). In particular, we utilize a mixture of transformer blocks which operate over audio and coarse (semantic) video sequences. We modify the MMDiT-X (Stability AI, 2024) layer architecture for our use-case, which, in addition to a joint attention operation applied to the sequence-wise concatenation of the query, key, and value projections of its inputs, uses an optional additional self-attention layer applied solely to the audio stream. We only apply this additional self-attention operation over the first $n$ blocks of the model. The output of these layers is then processed by additional standard single-sequence DiT blocks (Peebles & Xie, 2023). We utilize rotary positional embeddings (Su et al., 2023) in all transformer blocks.

**Representations.** Our model relies on three sources of conditioning information: low-resolution semantic visual features $v_s$, high-resolution visual synchronization features $v_{\text{syn}}$, and global text features $t_c$. $v_s$ is utilized both as a global feature (by pooling along the sequence axis), and in its original shape as an input to the multi-modal transformer blocks. $v_{\text{syn}}$ is used in sequence-level adaLN (Xu et al., 2019) conditioning layers after undergoing a learned upsampling operation to align its length with that of the audio sequence. For semantic video features, we use an off-the-shelf frozen SigLIP 2 (Tschannen et al., 2025) encoder, which is trained by contrastively aligning visual embeddings produced by a ViT (Dosovitskiy et al., 2021) with the corresponding text embeddings. We extract visual features frame-wise, using a lower frame rate of 8 FPS, with the intention of providing rich semantic information to the model, without yet focusing on precise synchronization. This process yields a sequence of shape $(t_{v_s}, 1152)$ where $t_{v_s}$ is the number of original video frames at 8 FPS. Text features are calculated using the corresponding text encoder, which yields a single 1152-dimensional vector per sequence. We train a dedicated model to learn synchronization features, which produces embeddings of shape $(t_{v_{\text{syn}}}, 768)$ which are sampled at 24 FPS.

### 3.1 SYNCHRONIZATION FEATURES & CONTRASTIVE PRE-TRAINING

Recent V2A models have employed dedicated contrastively-learned synchronization features for conditioning (Cheng et al., 2025), mostly based on the Synchformer model (Iashin et al., 2024). While these models are generally able to capture audio-visual time alignment features well, they can suffer from particular failure modes where some actions within the video are not recognized as such, which leads to no corresponding sound being generated. We hypothesize that this is due to the limited amount of data seen during pre-training, which shifts data requirements to the diffusion training stage. For this reason, we train our own contrastive synchronization model based on a significantly larger pre-trained vision backbone (VideoPrism) (Zhao et al., 2025). VideoPrism itself is based on the ViViT (Arnab et al., 2021) architecture, which employs factorized attention for the temporal and spatial axes.

We use the audio-spectrogram transformer model (AST) (Gong et al., 2021) as the audio encoder, which operates on mel-spectrogram features extracted from audio snippets with a sampling rate of 16 kHz. Following previous literature (Iashin et al., 2024), we use 128-dimensional mel features extracted from 25 ms audio segments with a hop size of 10 ms. We initialize this model with pretrained AST weights (Gong et al., 2021). The entire audio backbone is unfrozen and fine-tuned during training, while we add additional trainable layers and a MAP head (Zhai et al., 2022) on top of the otherwise frozen VideoPrism encoder. During training, both encoders receive snippets of length 0.667 seconds

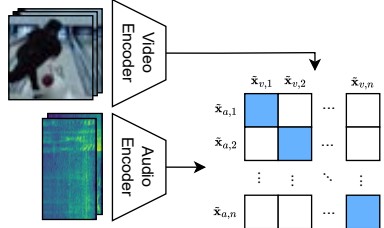

Figure 2: Contrastive alignment process. Groups of subsequent video frames are aligned with the corresponding audio snippets. Only patches corresponding to the same time range in the same batch are counted as positives.

of their respective modality (see Figure 2). The output embedding sequences are then averaged over their respective temporal axes to provide a single token for the input snippet for both modalities, $\tilde{\mathbf{x}}_v$ and $\tilde{\mathbf{x}}_a$. We then align these two vectors using a SigLIP loss objective (Zhai et al., 2023):

$$\mathcal{L}_{\text{SigLIP}} = -\frac{1}{|\mathcal{B}|} \sum_{i=1}^{|\mathcal{B}|} \sum_{j=1}^{|\mathcal{B}|} \log \frac{1}{1 + e^{z_{ij}(-t\tilde{\mathbf{x}}_{v,i} \cdot \tilde{\mathbf{x}}_{a,j} + b)}} \tag{1}$$

where $z_{ij}$ is the label for a given video/audio pair, equal to 1 if they are paired (that is, correspond to the same time range from the same video) and -1 otherwise, $\mathcal{B}$ is the batch size, and $\tilde{\mathbf{x}}_{v,i}$ and $\tilde{\mathbf{x}}_{a,j}$ are the encoder outputs for video & audio at batch positions $i$ and $j$. $t$ and $b$ are learned temperature and bias parameters, respectively.

An important aspect of this contrastive training process is the choice of batch size. In most contrastive pretraining tasks, higher batch sizes are usually preferred due to the model being exposed to more hard negatives against which the loss is evaluated. When training for precise synchronization, however, a larger batch size will result in an effective decrease of the proportion of hard negatives in the batch (snippets from the same video are generally more difficult to differentiate than snippets from other videos due to their high semantic similarity), transforming the task from that of learning synchronization and precise time alignment, to that of a more general semantic alignment process. We find that large batch sizes are worse for the learning features relevant to synchronization, which reflects similar findings made in Synchformer (Iashin et al., 2024), where it is noted that increasing the batch size beyond 28 (two videos yielding 14 snippets each) did not improve performance.

## 3.2 Masked Flow Matching

In order to enable audio-conditioned generation, we utilize a modified flow matching objective modified for in- and outpainting, using a similar approach as Chen et al. (2025). We do so with the intent of providing the model with the ability to seamlessly extend a provided ground-truth audio, enabling chunked long-form generation.

During training, we randomly mask a subset of the sequences in each batch with a given probability, with the mask spanning a contiguous range from 5 ms to 2.5 s. The masked subsequence contains the clean audio, and is thus not subject to the noising process resulting from flow matching training. Accordingly, the corresponding token positions are not included in the loss calculation, with the velocity prediction only being taken into account for noised tokens (ground-truth reference latents therefore do not contribute to the gradient). In order for the model to differentiate clean masked audio from the audio to be generated, we add two different learned tokens corresponding to masked and unmasked tokens to the relevant positions, following a technique employed in VampNet (Garcia et al., 2023). These masks are then also utilized during inference to mark the reference audio section (for details, see Appendix A.1). Beyond facilitating the conditioning on reference audio tracks, enabling users to generate audio that exhibits the spectral characteristics of a desired sample, this approach also enables in- and outpainting. The latter aspect is particularly relevant for long-form video-to-audio generation, which remains an issue in existing approaches. The vast majority of V2A models focus on the 8-12 second range, and experience a major reduction in performance when evaluated on longer sequences. Our model aims to circumvent this issue by continually extending samples generated on small snippets where synchronization remains high.

## 3.3 Shortcut Loss

We adopt a shortcut formulation (Frans et al., 2025) for our generative model to enable fast sampling without needing to retrain specifically for this purpose. To our knowledge, this is the first adoption of this formulation in the video-to-audio domain, as well as the larger field of general generative audio.

The input to the standard generative model used in flow matching is augmented with the desired step size $d$. This allows the model to generalize its velocity prediction to larger step sizes, essentially anticipating future curvature of the denoising path. Shortcut models are trained by enforcing a self-consistency property, in that one shortcut step must equal two shortcut steps of half that size (Frans et al., 2025). Training with the consistency property is done by setting a fraction (usually 25%) of the batch to self-consistency targets instead of the standard flow-matching denoising targets. Specifically, the training objective can be written as follows:

$$\mathcal{L}^{\mathrm{S}}(\theta) = E_{x_0, x_1, (t,d) \sim p(t,d)}[\underbrace{\|v_\theta(x_t, t, 0) - (x_1 - x_0)\|^2}_{\text{Flow-Matching}} + \underbrace{\|v_\theta(x_t, t, 2d) - v_{\text{target}}\|^2}_{\text{Self-Consistency}}], \quad (2)$$

where $x_0$ and $x_1$ represent samples from the noise and data distributions respectively, $t$ is the time step, $d$ is the sampled step size, and $x_t$ is the noisy latent (corresponding to $t$). $v_\theta$ represents the velocity prediction made by the model, and $v_{\text{target}}$ is the target velocity for the self-consistency loss.

## 4 EXPERIMENTS

We conduct experiments to evaluate our model's capabilities along multiple aspects: **1)** General capabilities under the standard V2A setting, i.e., text-conditioned audio generation for input clips from a variety of domains. This includes audio quality and synchronization. **2)** Audio-conditioned generation, in order to evaluate our model's ability to utilize a provided audio sample faithfully and realistically in a video clip. **3)** The capacity for few-step generation of high-quality results. **4)** Long-form capabilities, including the capacity to retain contextual aspects over longer horizons.

**Datasets.** In terms of audio-visual data, we use multiple datasets for training, including VGGSound (Chen et al., 2020) (∼550 h), and, to expand coverage, Moments-in-Time (Monfort et al., 2019) and a collection of in-the-wild YouTube videos obtained from Panda70M (Chen et al., 2024). While VGGSound provides good audio-visual alignment, we filter the rest of our collected data using our audio-visual synchronization model, where only those samples whose overall temporal and global semantic alignment exceed a relative threshold are kept. In total, this process results in ∼900 hours of audio-visual training data. For text supervision, we use AudioSetCaps captions for VGGSound (Bai et al., 2024), as well as WavCaps (Mei et al., 2024) and a high-fidelity studio Foley set for audio-text data. The total amount of audio-text data is thus ∼8.5k hours. Samples are cropped to a maximum length of 15 seconds. Evaluations are conducted on a test set composed of variable-length videos of up to 30 seconds, which contains samples from a holdout set of in-the-wild videos, the VGGSound test set, and UnAV-100 (Geng et al., 2023).

**Evaluation Metrics.** We assess the quality of our model using multiple metrics to measure audio similarity in learned representational spaces (distribution matching), overall quality, semantic & temporal alignment, and human preference. In particular, we compute the Fréchet Distance (Kilgour et al., 2019) using VGGish (Hershey et al., 2017) as the embedding model ($\text{FAD}_{\text{VGG}}$), and the Kullback-Leibler divergence using PaSST (Koutini et al., 2022) and PANNs (Kong et al., 2020). General audio quality is evaluated using the Inception Score (IS) (Salimans et al., 2016) based on PANNs, which is a reference-free measure to assess the clarity & diversity of generated samples. Semantic alignment between video and audio is measured using ImageBind (Girdhar et al., 2023) following the approach used in MMAudio and V-AURA (Cheng et al., 2025; Viertola et al., 2024). Features are extracted from the audio track and subsampled video frames, after which their embedding's average cosine similarity is calculated. Temporal alignment is evaluated using SynchFormer (Iashin et al., 2024) (DeSync), following the approach of MMAudio. This metric produces a misalignment value (in seconds) with a resolution of 0.2 seconds.

We also conduct a human listening study in which listeners are tasked with rating 10 videos (chosen for thematic diversity and to cover a range of sounds) from our test set along the axes of overall quality, temporal synchronization, and audio fidelity using a 5-point MOS scale. Audio tracks are shuffled blindly between trials (see Appendix A.3). The listening study was conducted with 18 qualified participants and included samples generated by our model, MMAudio (Cheng et al., 2025), and FoleyCrafter (Zhang et al., 2024).

### 4.1 RESULTS

Table 1 summarizes our model's performance on our test set against existing state-of-the-art models. It should be noted that this evaluation is conducted on mixed-length videos, as opposed to clips of fixed durations. SALSA-V surpasses previous approaches particularly in terms of synchronization, which is reflected in our human evaluation and in the DeSync metric. We also reach competitive scores in terms of distribution matching metrics, demonstrating the best $\text{FAD}_{\text{VGG}}$ and $\text{KL}_{\text{PANNs}}$ scores, while AudioX (Tian et al., 2025) performs best in terms of $\text{KL}_{\text{PANNs}}$. With regards

Table 1: Comparison with prior work. Left: objective metrics. Right: human evaluation (higher is better). "Sync." refers to synchronization (i.e., how well video frames and audio align). SALSA-V outperforms previous approaches on both objective (DeSync) and subjective human evaluation, highlighting its ability to better generate aligned audio.

| Method | Params | Objective metrics | | | | | | Human evaluation | | |
|--------|--------|------------------|--|--|--|--|--|----------------|--|--|
| | | $FAD_{VGG}\downarrow$ | $KL_{PANNs}\downarrow$ | $KL_{PaSST}\downarrow$ | IS↑ | IB↑ | DeSync↓ | Overall | Quality | Sync. |
| Frieren | 159M | 1.42 | 2.61 | 2.55 | 13.20 | 23.64 | 0.981 | - | - | - |
| V-AURA | 695M | 2.93 | 2.33 | 1.98 | 9.87 | 27.29 | 0.696 | - | - | - |
| LoVA | 1.06B | 1.79 | 2.14 | 2.06 | 16.86 | 27.95 | 1.205 | - | - | - |
| AudioX | 1.17B | 1.11 | **1.70** | **1.63** | 18.05 | 26.57 | 0.862 | - | - | - |
| FoleyCrafter | 1.22B | 2.43 | 2.21 | 2.15 | 16.39 | 26.37 | 1.319 | 2.78 | 2.46 | 2.37 |
| MMAudio | 1.03B | 1.12 | 1.77 | 1.72 | **18.13** | 32.89 | 0.521 | 3.29 | **3.16** | 2.92 |
| SALSA-V | 643M | **1.07** | 1.81 | **1.63** | 17.85 | **33.76** | **0.497** | **3.43** | 2.96 | **3.52** |

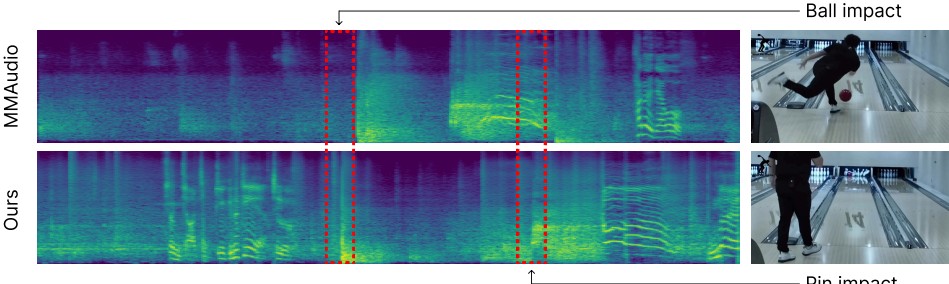

Figure 3: Qualitative example showing Mel-spectrogram of a generated audio clip with marked impacts, comparing SALSA-V with MMAudio. We observe that SALSA-V is able to better predict event alignment compared to previous methods.

to reference-free audio quality as measured by IS, and semantic alignment (IB) both our model and MMAudio achieve close scores. Human rating scores for overall quality are closer, with SALSA-V leading with a value of 3.43 versus MMAudio's 3.29.

We visualize the audio generated by our model on a representative sample from our validation set in Figure 3, which displays the generated mel-spectrogram. The sample is chosen for the marked impact sounds created by the bowling ball hitting first the track and then the pins. We find that SALSA-V is able to accurately pinpoint said events with high precision while also capturing the general duration of a given impact sound (e.g., the pin hit results in a more sustained sound, while the ball hitting the track has a fast attack and shorter duration). Moreover, our model retains the ability to generate appropriate background sounds during periods of limited video activity, as illustrated by the sections of human speech generated at the start and end of the segment.

Table 2: Ablation study on model size and diffusion variant. We report metrics of differently-sized model variants trained with either a default flow-matching loss or a shortcut objective. Utilizing the shortcut training objective does not reduce performance compared to flow matching. All models are evaluated at 32 sampling steps.

| Variant | Params | $FAD_{VGG}\downarrow$ | $KL_{PANNs}\downarrow$ | $KL_{PaSST}\downarrow$ | IS↑ | IB↑ | DeSync↓ |
|---------|--------|---------------------|----------------------|---------------------|-----|-----|---------|
| Flow Matching | 347M | 1.09 | 1.87 | 1.71 | 18.01 | 31.21 | 0.504 |
| Flow Matching | 643M | 1.10 | **1.80** | **1.63** | 17.91 | 33.71 | 0.499 |
| Shortcut | 347M | 1.12 | 1.85 | 1.68 | **18.03** | 31.09 | 0.501 |
| Shortcut | 643M | **1.07** | 1.81 | **1.63** | 17.85 | **33.76** | **0.497** |

Table 3: Comparison of metrics of Frieren (reflow) and SALSA-V at 4 sampling steps. SALSA-V demonstrates better performance in the few-step setting across all metrics.

| Method | Steps | $FAD_{VGG}\downarrow$ | $KL_{PANNs}\downarrow$ | $KL_{PaSST}\downarrow$ | IS↑ | IB↑ | DeSync↓ |
|---|---|---|---|---|---|---|---|
| Frieren (reflow) | 4 | 2.17 | 2.54 | 2.52 | 9.46 | 22.10 | 0.917 |
| SALSA-V | 4 | **1.19** | **1.83** | **1.67** | **14.57** | **31.18** | **0.536** |

**Few-Step Generation.** In order to test the few-step generation capabilities of SALSA-V, we compare the samples generated in the small-sampling-step regime with those of MMAudio (Cheng et al., 2025). The results of this analysis are shown in Figure 5, which includes FAD as a measure of semantic similarity to reference audio, and DeSync to measure synchronization. While MMAudio's performance decreases significantly across every metric as the number of sampling steps decreases, our model is able to stay competitive in the minimal sampling step regime, incurring almost no performance reduction at as little as 8 sampling steps in both metrics. It should be noted that our synchronization performance in particular is robust across all settings, as most impact sounds are already present after a few sampling steps, with the difference from results at a larger number of sampling steps being mainly composed of high-frequency textural details (see Figure 8). We also compare the performance of our shortcut model against an identical variant trained with a vanilla flow-matching loss. As can be seen in Table 2, we incur no performance reduction compared to the flow-matching variant at the full number of sampling steps (32 in this case), which indicates that the shortcut model training procedure does not compromise the model's ability to produce high-fidelity samples when needed. This stands in contrast to other approaches commonly used to reduce the number of sampling steps of flow matching and diffusion models, which usually entail fundamentally altering the learned velocity field in a post-training stage (e.g. the *reflow* procedure for rectified flow models (Liu et al., 2022)), which can harm fidelity at a larger number of sampling steps compared to the original model version.

To evaluate our model's ability in the few-step setting, we conduct an evaluation against Frieren (Wang et al., 2025) (using the *reflow* checkpoint, which is optimized for this task), with both models evaluated at 4 sampling steps. The results of this comparison can be seen in Table 3. SALSA-V displays better performance in all objective metrics in the few-step setting as well.

We assess the time taken by different models to generate a single 10-second sample in Table 4, evaluating dedicated few-step models in the full- and few-step sampling setting. SALSA-V exhibits competitive durations for high-quality samples, especially in the few-step regime. All models are executed on an NVIDIA A100 GPU at a batch size of 1.

**Conditional & Long-Form Generation.** Conditionally extended audio samples generally retain the spectral characteristics of their reference audio sequence, as qualitatively displayed in Figure 4, which uses a reference sample from our validation set. It can be seen that the extended sample faithfully transfers audio characteristics from the conditioning section, while still responding to unseen

Table 4: Generation efficiency of different models, measuring the time taken to generate a 10-second sample using a batch size of 1. SALSA-V is able to generate higher-quality results at 4 sampling steps, resulting in higher generation speeds than most competing models.

| Method | Params | Time (s) |
|---|---|---|
| V-AURA | 695M | 52.77 |
| FoleyCrafter | 1.22B | 5.53 |
| Frieren (25 steps) | 159M | 0.52 |
| Frieren (4 steps) | 159M | 0.12 |
| MMAudio | 1.03B | 6.49 |
| AudioX | 1.17B | 14.76 |
| LoVA | 1.06B | 8.45 |
| SALSA-V (32 steps) | 643M | 4.84 |
| SALSA-V (4 steps) | 643M | 0.71 |

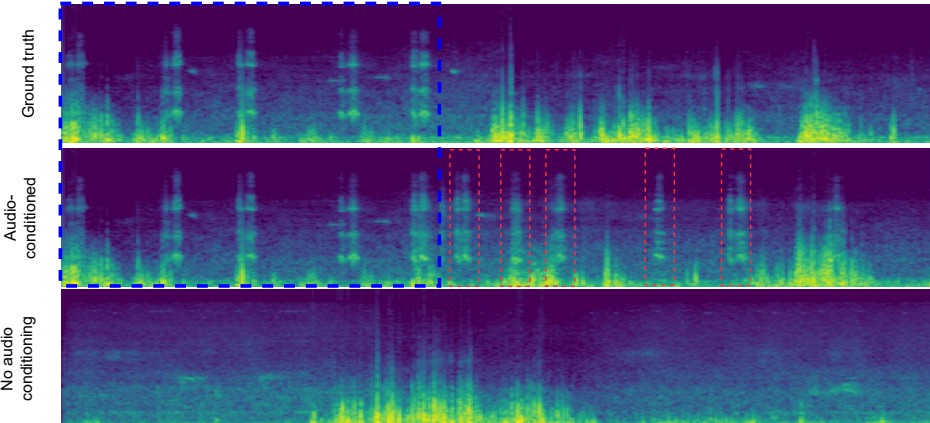

Figure 4: Qualitative example of outpainting capabilities of SALSA-V. The video used for this example contains a turkey gobbling (with its appropriate sound). The dashed blue region is given as context. The mel-spectrograms visualize a generation with and without audio conditioning. With audio conditioning, the model uses the characteristic sound present in the first 2 seconds and performs outpainting beyond that point (outside of the blue box). Both samples use the ground-truth video frames as visual information. With conditioning, the characteristic sound is utilized (the bird's call, highlighted by red boxes) and other spectral features, such as the frequency top-end, are also better-preserved.

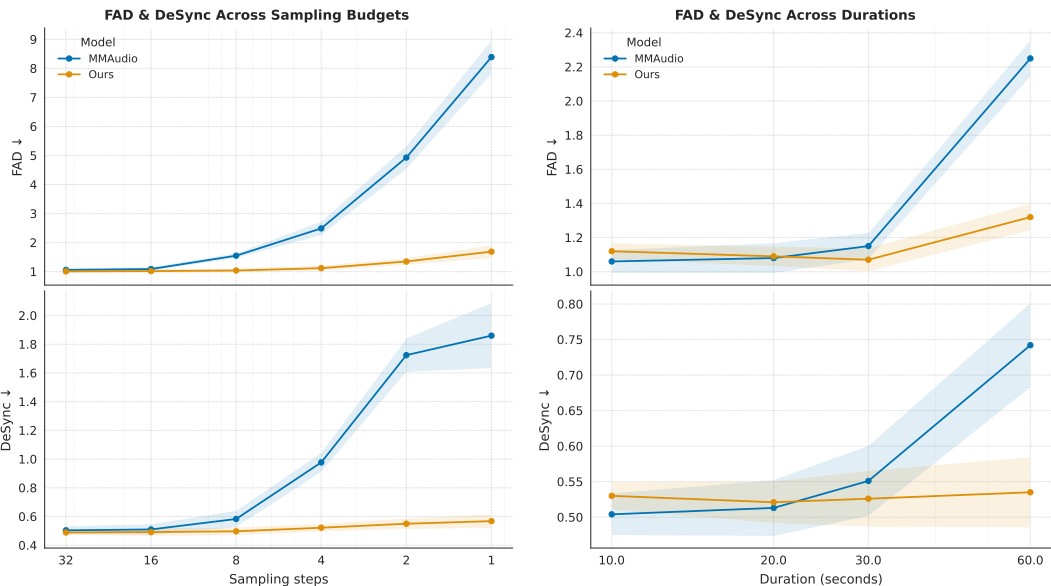

Figure 5: FAD and DeSync across different sampling steps (left) and generation lengths (right). SALSA-V is able to maintain generation quality with fewer sampling steps, and outperforms previous methods in long-form audio generation.

video events, as is evident by the increased frequency of repetition towards the end of the sample. Our model is able to accurately reuse a sample in the section to be generated, preserving both momentary effects and longer-range global frequency characteristics. We quantify these capabilities in Appendix A.2.

We investigate SALSA-V's long-form generation capabilities in terms of distribution matching and synchronization performance by calculating FAD and DeSync metrics for increasing generation lengths by using a set of long-form examples from our test set, with both models being evaluated on samples of a common, fixed duration. The corresponding results are displayed in Figure 5. After

Table 5: Long-form comparison between SALSA-V and LoVA on 30-second samples. SALSA-V displays superior results in all objective metrics.

| Method | $FAD_{VGG}\downarrow$ | $KL_{PANNs}\downarrow$ | $KL_{PaSST}\downarrow$ | IS↑ | IB↑ | DeSync↓ |
|--------|------|-------|-------|------|------|------|
| LoVA | 1.88 | 2.01 | 1.96 | 14.72 | 26.6 | 1.225 |
| SALSA-V | **1.09** | **1.73** | **1.71** | **16.34** | **31.88** | **0.526** |

generation lengths of approximately 20 seconds, MMAudio's performance begins to suffer, showing a reduction in both FAD and DeSync. By comparison, our model does not face these constraints, as we are able to repeatedly prepend the model's previous generation, in this case using a 0.5-second long reference section to generate 9.5 new seconds of audio. MMAudio instead processes the full sequence at once. We hypothesize that this improves MMAudio's performance in this comparison up to the 20-second range, as it has full access to the sequence, while we force our model to rely on a 10 second context in this setting as well. Beyond this range, MMAudio receives sequences with positional embeddings unseen during its training process, which explains the sudden decrease in synchronization performance and overall quality.

We conduct an evaluation of SALSA-V's long-form generation capabilities on 30-second samples against LoVA (Cheng et al., 2024), the results of which can be seen in Table 5. SALSA-V exhibits superior long-form generation capabilities in all objective metrics, maintaining particularly high scores in terms of synchronization and audio-visual alignment.

**Limitations.** Despite strong synchronization performance and competitive scores in terms of audiovisual alignment, our human evaluation study (Table 1) reveals a gap in subjective audio quality compared to MMAudio. Given the significantly smaller parameter count of our model (643M vs. 1.03B parameters for MMAudio), it is reasonable to hypothesize that scaling our model further could close or surpass this gap, further improving subjective quality. Future investigations could thus focus on larger-scale models, or techniques such as model distillation to enhance perceptual audio quality without proportionally increasing computational costs.

In addition, our current mechanism for long-form generation relies on the progressive outpainting of a prepended reference sample. This approach is mostly usable for single-shot videos without cuts between scenes displaying considerably different events, as the model will attempt to carry over the previous sound characteristics to this new section. However, this is mostly an issue when relying on automatic progressive extension without manual intervention; in practice, users can take steps such as re-seeding with a new generation or tuning the guidance value for audio conditioning to largely alleviate this issue.

## 5 CONCLUSION

We introduce SALSA-V, a video-to-audio model that demonstrates significant improvements over baselines in several key aspects. SALSA-V is particularly effective at synchronization with video content and overall quality at increased generation lengths while retaining sampling efficiency. Through a masked training objective, our model is able to jointly support audio conditioning and outpainting, which enables seamless long-form generations. The effectiveness of this approach is underscored by the quality and synchronization maintained over significantly extended durations. Furthermore, through the incorporation of a shortcut loss objective, SALSA-V supports few-step sampling without sacrificing sample fidelity at higher step counts or requiring dedicated fine-tuning procedures. Our experiments demonstrate that this training objective equips our model with the ability to generate high-quality audio samples even with a significantly reduced number of sampling steps. This aspect directly addresses one of the main limitations of conventional diffusion models in the video-to-audio domain, where a large number of sampling steps restrict their practical utility in real-time or resource-constrained applications.

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

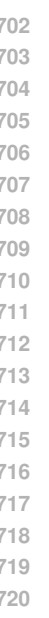

**SWSD & FAD Across Prepend Durations**

Figure 6: SWSD and FAD over increasing audio conditioning length. A prepending duration of 0 indicates a generation without audio-conditioning. Only the non-prepended section is considered for all calculations. Both metrics show increased similarity to the ground-truth audio for longer conditioning durations, with SWSD showing a more consistent reduction in spectral distance.

## A   APPENDIX

### A.1   MASKED AUDIO INFERENCE

We utilize classifier-free-guidance in conjunction with audio conditioning during inference as such:

$$\hat{v}(x_t, c, t) = v_\theta(x_t, t) + w \cdot (v_\theta(\hat{x}_t, c, t) - v_\theta(x_t, t)) \tag{3}$$

where $w$ is the guidance scale, $c$ is a collection of conditioning signals (semantic visual, synchronization visual, and text in our case), $v_\theta$ is the model's velocity prediction, and $\hat{x}_t$ represents the sequence to be generated at step $t$ with masking applied. More precisely, $\hat{x}_t$ is composed of $x_t$ and $x_1$ (the unconditional latent sequence, and the ground-truth sequence, respectively) in the following manner:

$$\hat{x}_t = \left(x_t^{(1)}, \ldots, x_t^{(n)}\right) \parallel x_1 \parallel \left(x_t^{(n+|x_1|+1)}, \ldots, x_t^{(T)}\right)$$

where $n$ depends on the desired starting position of the reference sequence, and $T = |x_t|$ is the length of the audio sequence to be generated.

### A.2   SPECTRAL CHARACTERISTICS OF AUDIO CONDITIONING

In order to quantify our model's ability to faithfully utilize audio conditioning information, we evaluate two different metrics: the per-sample FAD between the ground truth reference and the generated result, and a spectral difference metric intended to capture finer-grained frequency characteristics. This latter metric is intended to show whether specific characteristics of the frequency spectrum are preserved, while staying time-invariant (as we assess the quality of synchronization independently). We do this by treating the ground-truth and generated mel-spectrograms as probability distributions, calculating a Monte-Carlo estimate of the sliced Wasserstein distance (Rabin et al., 2012) between them. We refer to this metric as *Sliced-Wasserstein Spectral Distance* (SWSD).

To calculate SWSD, we first turn each signal (reference and generated) into a sequence of log-mel vectors (one per short-time frame). We then perform column-wise $\ell^1$ normalization, such that each

vector behaves like a probability mass function over mel bands. Those vectors are then treated as empirical samples from two high-dimensional distributions $\mathcal{P}_{\text{ref}}, \mathcal{P}_{\text{gen}} \subset \mathbb{R}^{\text{n\_mels}}$ between which we want to calculate a geometry-aware probability distance metric.

However, calculating the full Wasserstein distance between high-dimensional distributions is computationally infeasible, so we use a "sliced" variant, where the original high-dimensional samples are repeatedly projected on a number of randomly drawn unit vectors, between which we can efficiently calculate the 1-Wasserstein distance. This serves as an approximation of the full Wasserstein distance, with the estimate's accuracy increasing with the number of random unit vectors, which we set to 100 in our calculations. The mean of the 1-Wasserstein distances is used as our result. For details of our calculation, refer to Algorithm 1.

Both metrics are displayed in Figure 6, which shows the impact of audio-conditioned generation with different conditioning lengths on both FAD and SWSD. Values are calculated over a representative subset of audios with a maximum length of 10 seconds, which contain characteristic sounds corresponding to visual entities present in both the conditioning audio section, as well as in the section to be generated. Note that both metrics are only evaluated over these unseen sections (i.e., excluding the conditioning segment), in order to ensure a fair comparison. Our qualitative assessment is thus confirmed by both FAD and SWSD, which both show a significant decrease over the context length, before tapering off. While FAD is somewhat noisier, SWSD shows a steady decrease.

---

**Algorithm 1** Sliced-Wasserstein Spectral Distance

---

**Require:** Audio signals $x_{\text{ref}}, x_{\text{gen}} \in \mathbb{R}^n$, sampling rate $sr$
**Require:** Hyper-parameters: n_mels, hop_ms, win_ms, number of projections $K$, numerical $\varepsilon$
**Ensure:** Distance $d \in \mathbb{R}_{\geq 0}$
  1: **function** SLICEDWSMELDIST($x_{\text{ref}}, x_{\text{gen}}$)
  2:     $F_{\text{ref}} \leftarrow$ LOGMELFRAMES($x_{\text{ref}}$)                               $\triangleright T_{\text{ref}} \times$ n_mels
  3:     $F_{\text{gen}} \leftarrow$ LOGMELFRAMES($x_{\text{gen}}$)                               $\triangleright T_{\text{gen}} \times$ n_mels
  4:     **if** $T_{\text{ref}} \neq T_{\text{gen}}$ **then**
  5:         $T \leftarrow \min(T_{\text{ref}}, T_{\text{gen}})$
  6:         Randomly subsample each $F$ to $T$ rows
  7:     **else**
  8:         $T \leftarrow T_{\text{ref}}$                                    $\triangleright$ common frame count
  9:     **end if**
 10:    Sample $K$ random unit vectors $\{u_k\}_{k=1}^K \subset \mathbb{R}^{\text{n\_mels}}$
 11:    **for** $k = 1$ **to** $K$ **do**
 12:        $p_{\text{ref}} \leftarrow F_{\text{ref}} u_k$                                 $\triangleright$ project frames
 13:        $p_{\text{gen}} \leftarrow F_{\text{gen}} u_k$
 14:        Sort $p_{\text{ref}}$ and $p_{\text{gen}}$ ascending
 15:        $d_k \leftarrow \frac{1}{T} \sum_{i=1}^T |p_{\text{ref}}[i] - p_{\text{gen}}[i]|$
 16:    **end for**
 17:    **return** $d \leftarrow \frac{1}{K} \sum_{k=1}^K d_k$
 18: **end function**
 19: **function** LOGMELFRAMES($x$)
 20:    $hop \leftarrow \lfloor sr \cdot \text{hop\_ms}/1000 \rfloor$
 21:    $win \leftarrow \lfloor sr \cdot \text{win\_ms}/1000 \rfloor$
 22:    $S \leftarrow$ MELSPECTROGRAM($x, sr, \text{n\_mels}, win, hop$)
 23:    $S \leftarrow \log(S + \varepsilon)$
 24:    **for** each column $j$ **do** $S[:, j] \leftarrow S[:, j]/\sum_b S[b, j]$           $\triangleright \ell^1$ normalisation
 25:    **return** $S^\mathsf{T}$                                      $\triangleright$ shape $(T, \text{n\_mels})$
 26: **end function**

---

A.3 HUMAN EVALUATION

We develop an online interface to facilitate blind and randomized trials for our human listening evaluation. Figure 7 shows an example trial page. Listeners are tasked with rating 3 different tracks for a given video along 3 different aspects, each using a 5-point MOS scale:

- **Overall Fit**: How well the given audio track semantically corresponds to the video.

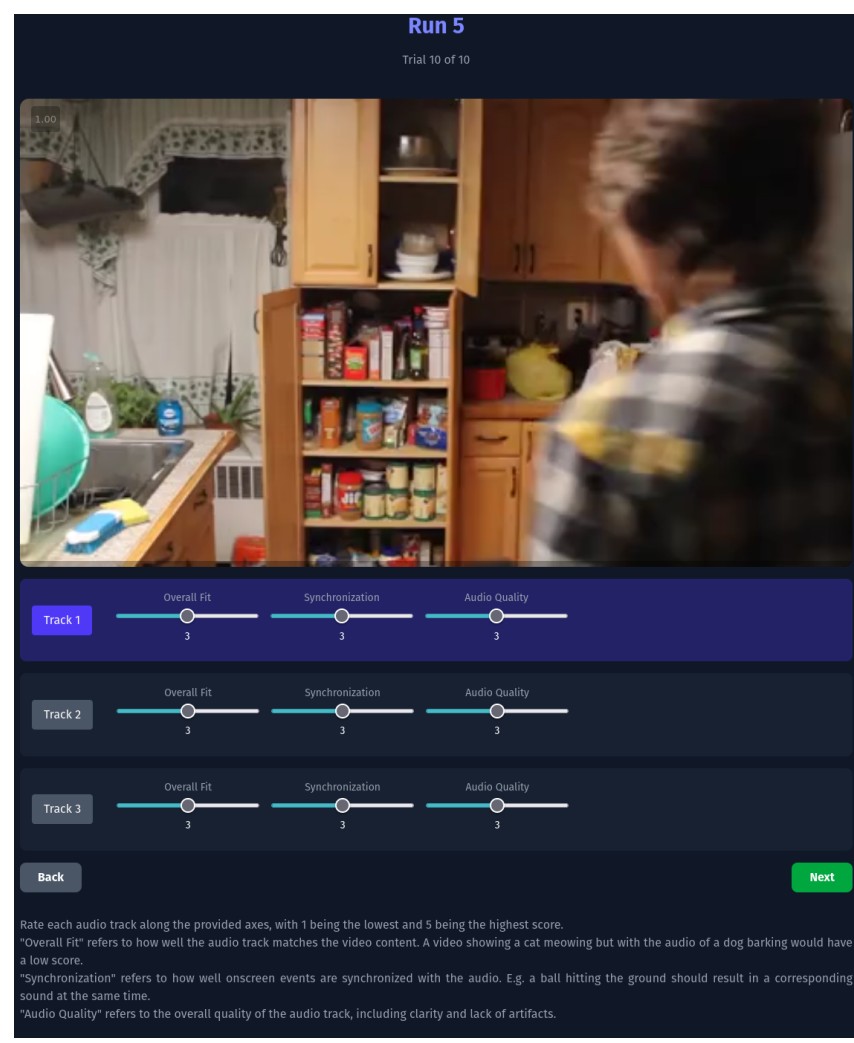

Figure 7: An example screenshot showing the interface of the listening test used for our human evaluation study. Audio track identifiers are randomized both between trials and runs.

- **Synchronization**: How precisely video events are synchronized to their corresponding sound.
- **Audio Quality**: General quality of the audio, encompassing aspects such as artifacts and clarity.

Each listener completes one test run, where a single run consists of 10 trials. In our case, this resulted in 180 individual scores. Particular care was taken to ensure that synchronization is maintained in the web interface when switching between audio tracks, as even small drifts over time can affect results.

### A.4 IMPLEMENTATION DETAILS

**Length Unification.** To unify the sequence lengths of our synchronization conditioning signal to the audio input, we employ a combination of cubic interpolation and a learned upsampling layer, yielding features that are frame-aligned to the audio latent sequence.

More precisely, resampled synchronization video features are calculated as follows:

$$\tilde{v}_{\text{syn}} = \text{LayerNorm}(\text{stopgrad}(\text{interp}(v_{\text{syn}}))) + \tanh(g) \cdot \text{ConvTranspose}(v_{\text{syn}}) \quad (4)$$

where $v_{\text{syn}}$ are the visual synchronization features at their original temporal resolution, $g$ is a learned gating value, "interp" denotes cubic interpolation, and "stopgrad" is the stop-gradient operation. We find that stopping the gradient from flowing through the interpolation path helps stabilize training early on and prevents the model from collapsing the learned upsampling path.

**Training & Evaluation Details.** We use a batch size of 30 for training of the contrastive audio-visual alignment model, which is done on VGGSound (Chen et al., 2020). We keep the VideoPrism (Zhao et al., 2025) backbone frozen while adding 4 additional spatial, and 2 temporal encoder layers (using the same underlying architecture as ViViT (Arnab et al., 2021)) which are trainable. Furthermore, we unfreeze the complete audio backbone and perform training at a learning rate of $3 \times 10^{-5}$. The new task layers are set to a learning rate of $1 \times 10^{-4}$. We use the AdamW (Loshchilov & Hutter, 2019) optimizer with a weight decay of $1 \times 10^{-4}$ and a cosine learning rate scheduler with 2000 warmup steps, starting at an initial learning rate of 0. All models are trained for 1M steps. We discard input samples with an audio/frame ratio outside of a 5% grace range centered around the value expected for our configuration ($\frac{16000 \text{ audio samples}}{24 \text{ frames}} \approx 666.67$). This is done to ensure precise synchronization between visual events and their corresponding sound.

The visual encoder outputs a tensor of shape $(t_v, s, d)$ per input snippet, where $t_v$, $s$, and $d$ are the temporal, spatial, and depth dimensions, respectively. The spatial dimension $s$ is reduced using a MAP head, yielding a $(t_v, d)$ tensor. This tensor is used as-is as the synchronization conditioning feature for the generative model, while additionally undergoing a pooling operation during the contrastive model training stage (yielding a single $d$-sized vector per input snippet for the loss calculation).

The audio encoder outputs a tensor of shape $(f, t_a, d)$, where $f$, $t_a$, and $d$ are the frequency, time, and depth dimensions. This tensor is processed using a similar aggregation operation as the visual output, but along the frequency dimension, which results in an output of shape $(t_a, d)$. After pooling along the time dimensions, the resulting vector is used in the contrastive loss calculation.

The main generative model is trained with a batch size of 300 and a consistency target ratio of 25%, using the Adam (Kingma & Ba, 2017) optimizer with a learning rate of $1 \times 10^{-4}$. Time steps for the noising process are sampled according to a log-normal distribution, and we use EMA with a target update rate of 0.999, training for 1M steps in total. All conditioning signals are dropped out with a probability of 0.1 during training, and 0.25 of the input sequences in a batch are modified by the audio masking process which affects a random contiguous subsection of the sequence with a randomly chosen size between 1 and 88 latent tokens. We use a CFG guidance value of 4.0 for our experiments while using author-recommended settings for all other models.

Our large model uses 6 multi-modal blocks where the first 3 include the optional self-attention operation over the audio sequence, followed by 6 additional single-modality DiT blocks. Our smaller model variant uses 4 multi-modal blocks, with the first 2 including separate self-attention, followed by 4 single-modality blocks.

All models are implemented in JAX (Bradbury et al., 2018) using the Flax (Heek et al., 2024) machine learning library and trained on TPU v4-8 nodes. Other relevant models used during training or preprocessing (AST (Gong et al., 2021), Stable-Audio VAE (Evans et al., 2024)) were converted to JAX.

## A.5 VAE TRAINING

We fine-tune the Stable-Audio VAE (Evans et al., 2024), increasing its sampling rate from 21.5 Hz to 43 Hz by replacing the first encoder and last decoder layers. The VAE is then fine-tuned from this state for 800k steps, using a batch size of 128. This training process was conducted on audio samples from VGGSound (Chen et al., 2020), AudioSet (Gemmeke et al., 2017), and our high-fidelity studio Foley dataset.

This was done both to increase the quality of the audio codec, as it is limited to a large degree by the downsampling ratio used in the encoder, and to align the training data distribution more closely with our intended application, as Stable-Audio VAE is trained on a significant amount of speech data (while we focus on general environmental sounds).

## A.6 ADDITIONAL RESULTS

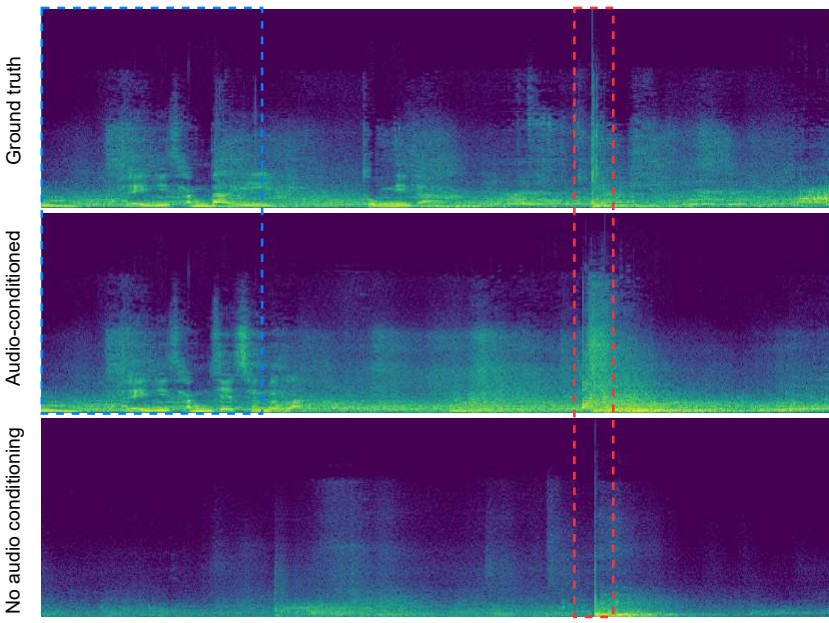

Figure 8: Spectrograms of samples for a 10-second video of a bird chirping generated at a different number of sampling steps (32, 8, and 1, i.e. single-shot generation) using our shortcut model. General audio structure is preserved among all settings, with the single-shot generation mostly losing fidelity in higher frequency ranges and short transients.

Figure 9: Spectrograms of samples for a 5-second video of a pronounced impact sound with visual correspondence (a baseball bat hitting a ball) with and without audio conditioning. The audio conditioning portion is in the blue box, while the impact is highlighted with red. As can be seen, audio-conditioned generations also remain responsive to visual cues (i.e. no over-reliance on ground-truth audio).

