# OpenReview forum: "SALSA-V: Shortcut-Augmented Long-form Synchronized Audio from Videos"
_ICLR.cc/2026/Conference — Submitted to ICLR 2026_

### Official Review · Reviewer_gTEV · 2025-10-27

**Soundness:** 2
**Presentation:** 3
**Contribution:** 2
**Rating:** 2
**Confidence:** 4

**Summary:**

This paper presents a new video-to-audio model called SALSA-V that efficiently generates long-form audio given a target video. To perform long-form audio generation, the proposed method iteratively outpaints the previously generated audio, and for that purpose, the model is trained with masked ground-truth audio as conditional input. For efficient generation, the training objective includes a shortcut loss, which enables few-step generation during inference. The experimental results show that the proposed model outperforms existing models, especially when the duration of the target video is long or the number of sampling steps is small.

**Strengths:**

- The proposed method is simple and should be easy to implement.
- The advantage of the proposed method over MMAudio is significant when the duration of the target video is long or the number of sampling steps is small.

**Weaknesses:**

- The novelty of the methodology is marginal.
  - The main modifications from MMAudio are training with masked audio and the use of shortcut loss, and both techniques appear to be adopted in a straightforward manner.
  - It would be beneficial to clearly describe the particular challenges in applying these techniques to video-to-audio models and how the proposed method addresses them. Alternatively, the authors can mention any empirical insights that could be helpful for future studies in the field of video-to-audio generation.
- The comparison in the experiments needs additional baselines.
  - For efficient generation, Frieren (or applying reflow to the proposed model with a flow matching formulation), as mentioned in Section 2.3, would be a good baseline.
  - For long-form generation, V-AURA (mentioned in Section 2.1) and LoVA (mentioned in Section 2.4) could be used as baselines.
- The benefit of audio-conditional generation in SALSA-V is not clear in Figure 4.
  - I understand that some audio patterns in the conditional audio faithfully appear in the generated audio, but it seems that the ground-truth audio does not actually contain such patterns within the generated timeline. Thus, it is not clear if the conditional audio contributes to accurate audio generation.
- I could not access the demo page due to a timeout error.

**Questions:**

- Is there any reason why the authors did not try SALSA-V with 1B parameters?

---

> ### Author Response · Authors · 2025-11-20
>
> We thank the reviewer for their feedback. In the following we address each concern.
>
> > For efficient generation, Frieren (or applying reflow to the proposed model with a flow matching formulation), as mentioned in Section 2.3, would be a good baseline. For long-form generation, V-AURA (mentioned in Section 2.1) and LoVA (mentioned in Section 2.4) could be used as baselines.
>
> We have added additional comparisons with Frieren (non-reflowed at 25 steps, reflowed at 4 steps), LoVA, and V-AURA in the overall comparison, as well as in the long-form setting. We find that SALSA-V exhibits better performance in most metrics compared to previous methods, especially when taking runtime into consideration.
>
>
> Overall metrics:
> |                    | Params | FAD_VGG  | KL_PANNs | KL_PaSST | IS        | IB        | DeSync    | Runtime (s) |
> |--------------------|--------|----------|----------|----------|-----------|-----------|-----------|-------------|
> | FoleyCrafter       | 1.22B  |     2.43 |     2.21 |     2.15 |     16.39 |     26.37 |     1.319 |        5.53 |
> | Frieren (4 steps)  | 159M   |     2.17 |     2.54 |     2.52 |      9.46 |      22.1 |     0.917 |        0.12 |
> | Frieren (25 steps) | 159M   |     1.42 |     2.61 |     2.55 |      13.2 |     23.64 |     0.981 |        0.52 |
> | V-AURA             | 695M   |     2.93 |     2.33 |     1.98 |      9.87 |     27.29 |     0.696 |       52.77 |
> | LoVA               | 1.06B  |     1.79 |     2.14 |     2.06 |     16.86 |     27.95 |     1.205 |        8.45 |
> | MMAudio            | 1.03B  |     1.12 |     1.77 |     1.72 | **18.13** |     32.89 |     0.521 |        6.49 |
> | AudioX             | 1.17B  |     1.11 |  **1.7** | **1.63** |     18.05 |     26.57 |     0.862 |       14.76 |
> | SALSA-V (4 steps)  | 643M   |     1.19 |     1.83 |     1.67 |     14.57 |     31.18 |     0.536 |        0.71 |
> | SALSA-V (32 steps) | 643M   | **1.07** |     1.81 | **1.63** |     17.85 | **33.76** | **0.497** |        4.84 |
>
> Long-form comparison against LoVA on 30-second samples:
>
>
> |         | FAD_VGG  | KL_PANNs | KL_PaSST | IS        | IB        | DeSync    |
> |---------|----------|----------|----------|-----------|-----------|-----------|
> | LoVA    |     1.88 |     2.01 |     1.96 |     14.72 |      26.6 |     1.225 |
> | SALSA-V | **1.09** | **1.73** | **1.71** | **16.34** | **31.88** | **0.526** |
>
> > I understand that some audio patterns in the conditional audio faithfully appear in the generated audio, but it seems that the ground-truth audio does not actually contain such patterns within the generated timeline. Thus, it is not clear if the conditional audio contributes to accurate audio generation.
>
> Audio-conditional generation is not primarily intended to increase the accuracy of audio generation, but to provide additional options for users, e.g. by selecting a specific audio portion as a sample to reuse, or to facilitate inpainting in an existing audio track. Due to this, audio-conditioned results might not necessarily reflect future changes in the conditioning audio, but rather the ability to faithfully reuse a certain sound in instances where it would otherwise not appear.
>
> > I could not access the demo page due to a timeout error.
>
> We thank the reviewer for informing us of this issue. We have created a private repository to host the samples in addition to the anonymized link, which can be accessed here: https://salsav.pages.dev/
>
> In case there are still issues accessing or playing the demo videos, we kindly ask the reviewer to inform us of this.
>
> > Is there any reason why the authors did not try 1B parameters
>
> The main claims in our manuscript are already supported by the empirical results observed for our 643M parameter model, showing that we outperform previous state-of-the-art methods in long-form and few-step generation even at a lower parameter count. Therefore, we did not further investigate scaling behavior beyond the increase from 347M to 643M parameters.

---

> > ### Comment · Reviewer_gTEV · 2025-11-25
> >
> > Thanks for providing the additional experimental results and also for fixing the demo page issue. I can now successfully access to the demo page.
> >
> > I have two follow-up questions:
> >
> > > Audio-conditional generation is not primarily intended to increase the accuracy of audio generation, but to provide additional options for users, e.g. by selecting a specific audio portion as a sample to reuse, or to facilitate inpainting in an existing audio track. Due to this, audio-conditioned results might not necessarily reflect future changes in the conditioning audio, but rather the ability to faithfully reuse a certain sound in instances where it would otherwise not appear.
> >
> > While I understand the authors’ argument here, I still wonder whether the example shown in Fig. 4 is appropriate for demonstrating the advantage of the proposed method.
> >
> > > The main claims in our manuscript are already supported by the empirical results observed for our 643M parameter model, showing that we outperform previous state-of-the-art methods in long-form and few-step generation even at a lower parameter count. Therefore, we did not further investigate scaling behavior beyond the increase from 347M to 643M parameters.
> >
> > I am not fully convinced by this explanation for two reasons. First, the authors suggest that the smaller model size may explain why SALSA-V did not outperform MMAudio in subjective audio quality. This could be directly tested by evaluating SALSA-V with a 1B-parameter model. Second, since SALSA-V is based on MMAudio, it would be valuable to assess its scalability at the same parameter scale as MMAudio to ensure that the added modules do not negatively impact the model’s scalability.

---

### Official Review · Reviewer_7VCU · 2025-10-28

**Soundness:** 2
**Presentation:** 1
**Contribution:** 2
**Rating:** 4
**Confidence:** 3

**Summary:**

This paper presents SALSA-V, a shortcut-augmented latent flow matching model for video-to-audio generation that achieves high-fidelity and temporally synchronized audio in just a few sampling steps. By combining masked training for audio conditioning and outpainting with contrastively trained synchronization features, SALSA-V enables both efficient short-form and stable long-form audio generation without distillation.

**Strengths:**

**1. Enables Long-Form Audio Generation**

The model supports audio-conditioned outpainting, allowing stable and coherent generation of extended audio sequences (30 s+) from video inputs.

**2. Improved Audio-Visual Synchronization**

A contrastively trained synchronization encoder provides precise temporal alignment between visual motion and audio events, achieving SOTA synchronization performance.

**3. Efficient Few-Step Sampling**

The shortcut-augmented flow matching formulation reduces sampling steps to ≤ 8 without quality degradation.

**Weaknesses:**

**1. Low Readability and Clarity**

The paper’s presentation is occasionally hard to follow.

**2. Inconsistency between Quantitative Results (Fig. 5 vs. Table 1)**

There appears to be a mismatch between the 10s performance trend in Figure 5 and the metrics in Table 1.

**3. Insufficient Analysis of Inference Efficiency**

While the paper emphasizes few-step generation, there is no thorough empirical analysis of inference speed. More concrete runtime results would strengthen the efficiency claims.

**Questions:**

**1. Clarification on Feature Roles and Figure References**

In the Method section, the paper sequentially introduces the VAE encoder, visual-text representation, and synchronization feature.
However, it is unclear how each of these corresponds to the components shown in Figure 1, and what specific roles they play in the generation pipeline. A clearer mapping between the described features and their positions in Figure 1 would be helpful.
In addition, Figure 5 is never explicitly referenced in the main text—please consider mentioning and explaining it within the corresponding experimental section.

**2. Inconsistency between Figure 5 and Table 1 Results**

In Figure 5, the proposed model shows lower performance than MMAudio for 10-second generation, whereas Table 1 indicates a different trend. Could you clarify whether these results are based on different experimental setups or evaluation protocols, and explain what accounts for the discrepancy?

**3. Quantitative Analysis of Sampling Efficiency**

The paper highlights sampling efficiency as a major advantage of SALSA-V, but does not provide concrete runtime comparisons.
Could the authors include or discuss actual inference speed measurements (e.g., seconds per 10-second clip, GPU type, batch size), and how they compare to other models?

---

> ### Author Response · Authors · 2025-11-20
>
> We thank the reviewer for their feedback. In the following we address each concern.
>
> > A clearer mapping between the described features and their positions in Figure 1 would be helpful. In addition, Figure 5 is never explicitly referenced in the main text—please consider mentioning and explaining it within the corresponding experimental section.
>
> We have updated Figure 1 to highlight the correspondence between the components discussed in the methodology and how they relate to the overall generation pipeline.
> We thank the reviewer for highlighting the issue of Figure 5 not being referenced in the text. This was due to a LaTeX formatting issue, which rendered all references to "Figure 5" as "Section 4.1". We have updated the manuscript to fix this.
>
> > In Figure 5, the proposed model shows lower performance than MMAudio for 10-second generation, whereas Table 1 indicates a different trend. Could you clarify whether these results are based on different experimental setups or evaluation protocols, and explain what accounts for the discrepancy?
>
> Table 1 is evaluated on mixed-length samples from the entire test set, while Figure 5 only evaluates on the duration indicated on the x-axis. Due to this, evaluation results differ between the two. We have updated the manuscript to clarify this.
>
> > The paper highlights sampling efficiency as a major advantage of SALSA-V, but does not provide concrete runtime comparisons. Could the authors include or discuss actual inference speed measurements (e.g., seconds per 10-second clip, GPU type, batch size), and how they compare to other models?
>
> We have included an additional comparative analysis of sampling speeds of other models, focusing on the overall runtime required to generate a single 10-second sample (batch size of 1, measured on an A100 GPU).
>
> While generation speeds mostly scale with parameter count, SALSA-V compares favourably against similarly-sized models, especially in the few-step regime, being able to generate a high-quality 10-second sample in 0.71 seconds in this setting.
>
> |                    | Params | FAD_VGG  | KL_PANNs | KL_PaSST | IS        | IB        | DeSync    | Runtime (s) |
> |--------------------|--------|----------|----------|----------|-----------|-----------|-----------|-------------|
> | FoleyCrafter       | 1.22B  |     2.43 |     2.21 |     2.15 |     16.39 |     26.37 |     1.319 |        5.53 |
> | Frieren (4 steps)  | 159M   |     2.17 |     2.54 |     2.52 |      9.46 |      22.1 |     0.917 |        0.12 |
> | Frieren (25 steps) | 159M   |     1.42 |     2.61 |     2.55 |      13.2 |     23.64 |     0.981 |        0.52 |
> | V-AURA             | 695M   |     2.93 |     2.33 |     1.98 |      9.87 |     27.29 |     0.696 |       52.77 |
> | LoVA               | 1.06B  |     1.79 |     2.14 |     2.06 |     16.86 |     27.95 |     1.205 |        8.45 |
> | MMAudio            | 1.03B  |     1.12 |     1.77 |     1.72 | **18.13** |     32.89 |     0.521 |        6.49 |
> | AudioX             | 1.17B  |     1.11 |  **1.7** | **1.63** |     18.05 |     26.57 |     0.862 |       14.76 |
> | SALSA-V (4 steps)  | 643M   |     1.19 |     1.83 |     1.67 |     14.57 |     31.18 |     0.536 |        0.71 |
> | SALSA-V (32 steps) | 643M   | **1.07** |     1.81 | **1.63** |     17.85 | **33.76** | **0.497** |        4.84 |

---

> > ### Comment · Reviewer_7VCU · 2025-11-26
> >
> > Thank you for your efforts in addressing my concerns. While I appreciate the clarifications, I still have some reservations regarding the overall quality of the paper, particularly the issue raised in Weakness 3 by reviewer jLCz.
> > Therefore, I will maintain my original scores.

---

### Official Review · Reviewer_jLCz · 2025-10-30

**Soundness:** 4
**Presentation:** 3
**Contribution:** 1
**Rating:** 4
**Confidence:** 5

**Summary:**

This paper introduces SALSA-V, a model for video-to-audio (V2A) generation, designed to synthesize high-fidelity, long-form audio that is precisely synchronized with video content. The authors propose three main contributions: (1) a shortcut-augmented training objective to enable high-quality audio generation in very few sampling steps; (2) a masked flow matching approach that allows the model to perform audio-conditioned generation and outpainting, thereby enabling the creation of audio for long-form videos through iterative extension; and (3) a new contrastive audio-visual synchronization model built upon a strong pre-trained vision backbone to yield high-resolution alignment features.

**Strengths:**

*   **Clear Presentation and Structured Evaluation**: The paper is clearly written, with a well-defined problem motivation and a structured presentation of its methods. The evaluation is methodical, employing a range of established objective metrics alongside a human listening study.

*   **Focus on Practical V2A Challenges**: The work addresses relevant practical limitations in the video-to-audio (V2A) domain, particularly the efficiency of the sampling process and the synthesis of audio for longer-form videos. This focus is pertinent to improving the applicability of such generative models.

*   **Achieves Strong Temporal Synchronization**: A notable strength of the proposed model is its ability to generate tightly synchronized audio. The paper reports state-of-the-art results on the DeSync metric, and this quantitative improvement in temporal alignment is also corroborated by the human evaluation study.

**Weaknesses:**

The paper, while presenting a well-engineered system, suffers from several weaknesses that question the significance and novelty of its contribution.

1.  **Limited Novelty and Incremental Contribution**: The primary weakness of this work is its reliance on existing techniques across its entire pipeline, from the architectural framework and training methods to the experimental conclusions. This makes the overall contribution feel incremental rather than innovative.
    *   The model's core design, which combines semantic and high-resolution synchronization features for conditioning, directly follows the framework established by **MMAudio[1]**.
    *   The use of a masked training objective for audio conditioning is a known technique, with similar approaches having been explored in works like **AudioX[2]** and **MultiFoley[3]**.
    *   The key insight regarding the choice of batch size for training the synchronization model is also acknowledged to be a direct replication of findings from **Synchformer [4]**.
    While the integration of these parts is functional, the paper does not introduce a new fundamental concept, algorithm, or a significant insight to the field.

2.  **Insufficient Experimental Baseline and Missing Citations**: The paper's experimental comparison is narrow, failing to benchmark against a sufficient number of relevant contemporary models. This makes it difficult to accurately assess its performance in the broader context of the field.
    *   The main quantitative comparison (Table 1) only includes two other models, **FoleyCrafter** and **MMAudio**. Other highly relevant V2A methods, such as the autoregressive model **V-AURA [5]** and other diffusion-based models like **AudioX** and **Frieren[6]**, are not included in the benchmark.
    *   A more comprehensive comparison against a wider array of recent models is necessary to robustly support the claims of state-of-the-art performance.

3.  **Contradictory Results and Lack of Significant Improvement**: Several claims in the paper are not fully supported by its own results, and the overall improvement is not compelling.
    *   **Lower Subjective Audio Quality**: For a generative model, perceptual quality is paramount. The human evaluation in Table 1 shows that SALSA-V's **Audio Quality** score (2.96) is lower than the baseline MMAudio (3.16). This key result undermines the paper's claim of outperforming existing methods. The overall subjective improvement is not significant.
    *   **Potentially Flawed Long-Form Evaluation**: The paper compares its iterative, chunk-based generation with MMAudio's one-shot, full-sequence approach. As the models operate with different context windows and inference strategies, this direct comparison of metrics could be misleading.
    *   **Disconnect Between Main Contribution and SOTA Results**: The "shortcut loss" for few-step sampling is highlighted as a major contribution. However, the main results in Table 1, which establish the model's SOTA synchronization, are based on 32 sampling steps. The paper does not demonstrate that this claimed SOTA performance is retained in the few-step regime, thus disconnecting the novel claim from the primary comparative results.

4.  **Lack of Clarity in Experimental Details and Reproducibility Issues**: The description of the experimental setup lacks the necessary detail for reproducibility.
    *   The test set used for evaluation is vaguely described as a composition of "a holdout set of in-the-wild videos, the VGGSound test set, and UnAV-100". Without precise details on the composition, data splits, and preprocessing of this custom benchmark, the results are not verifiable or reproducible by the community.

[1] Cheng H K, Ishii M, Hayakawa A, et al. MMAudio: Taming Multimodal Joint Training for High-Quality Video-to-Audio Synthesis[C]//Proceedings of the Computer Vision and Pattern Recognition Conference. 2025: 28901-28911.

[2] Tian Z, Jin Y, Liu Z, et al. Audiox: Diffusion transformer for anything-to-audio generation[J]. arXiv preprint arXiv:2503.10522, 2025.

[3] Chen Z, Seetharaman P, Russell B, et al. Video-guided foley sound generation with multimodal controls[C]//Proceedings of the Computer Vision and Pattern Recognition Conference. 2025: 18770-18781.

[4] Iashin V, Xie W, Rahtu E, et al. Synchformer: Efficient synchronization from sparse cues[C]//ICASSP 2024-2024 IEEE International Conference on Acoustics, Speech and Signal Processing (ICASSP). IEEE, 2024: 5325-5329.

[5] Viertola I, Iashin V, Rahtu E. Temporally aligned audio for video with autoregression[C]//ICASSP 2025-2025 IEEE International Conference on Acoustics, Speech and Signal Processing (ICASSP). IEEE, 2025: 1-5.

[6] Wang Y, Guo W, Huang R, et al. Frieren: Efficient video-to-audio generation network with rectified flow matching[J]. Advances in Neural Information Processing Systems, 2024, 37: 128118-128138.

**Questions:**

1.  **On Novelty**: The paper's core components (architecture, masked training, batch size insights) appear to be adapted from prior work. Could the authors clarify the primary technical novelty beyond the successful integration of these known techniques?

2.  **On Experimental Baselines**: The experimental comparison is limited. Could the authors justify the exclusion of several recent and relevant baselines like V-AURA, AudioX, and Frieren, and perhaps provide a comparative analysis on key metrics?

3.  **On Few-Step Generation Performance**: The "shortcut loss" is a key claimed contribution, but SOTA results are shown at 32 steps. Could the authors provide key metrics (e.g., DeSync, FAD) for the 8-step generation case against baselines to demonstrate the practical effectiveness of this feature?

4.  **On Reproducibility**: The evaluation benchmark is vaguely described. For reproducibility, could the authors provide a precise composition and list of identifiers for their custom test set?

---

> ### Author Response · Authors · 2025-11-20
>
> We thank the reviewer for their feedback. In the following we address each concern.
>
> > Could the authors clarify the primary technical novelty beyond the successful integration of these known techniques?
>
> While the individual components that make up our proposed pipeline are known, their combination has not been explored before. SALSA-V is able to outperform all existing models in long-form video-to-audio generation, resulting in a new state-of-the-art. Therefore, we believe that our model serves as a valuable contribution to the community.
>
> > Could the authors justify the exclusion of several recent and relevant baselines like V-AURA, AudioX, and Frieren, and perhaps provide a comparative analysis on key metrics?
>
> We have extended our evaluation to include the comparison with V-AURA, AudioX, and Frieren. SALSA-V overall outperforms previous methods.
>
> |                    | Params | FAD_VGG  | KL_PANNs | KL_PaSST | IS        | IB        | DeSync    | Runtime (s) |
> |--------------------|--------|----------|----------|----------|-----------|-----------|-----------|-------------|
> | FoleyCrafter       | 1.22B  |     2.43 |     2.21 |     2.15 |     16.39 |     26.37 |     1.319 |        5.53 |
> | Frieren (4 steps)  | 159M   |     2.17 |     2.54 |     2.52 |      9.46 |      22.1 |     0.917 |        0.12 |
> | Frieren (25 steps) | 159M   |     1.42 |     2.61 |     2.55 |      13.2 |     23.64 |     0.981 |        0.52 |
> | V-AURA             | 695M   |     2.93 |     2.33 |     1.98 |      9.87 |     27.29 |     0.696 |       52.77 |
> | LoVA               | 1.06B  |     1.79 |     2.14 |     2.06 |     16.86 |     27.95 |     1.205 |        8.45 |
> | MMAudio            | 1.03B  |     1.12 |     1.77 |     1.72 | **18.13** |     32.89 |     0.521 |        6.49 |
> | AudioX             | 1.17B  |     1.11 |  **1.7** | **1.63** |     18.05 |     26.57 |     0.862 |       14.76 |
> | SALSA-V (4 steps)  | 643M   |     1.19 |     1.83 |     1.67 |     14.57 |     31.18 |     0.536 |        0.71 |
> | SALSA-V (32 steps) | 643M   | **1.07** |     1.81 | **1.63** |     17.85 | **33.76** | **0.497** |        4.84 |
>
> > The "shortcut loss" is a key claimed contribution, but SOTA results are shown at 32 steps. Could the authors provide key metrics (e.g., DeSync, FAD) for the 8-step generation case against baselines to demonstrate the practical effectiveness of this feature?
>
> In addition to Figure 5, which displays our model's performance in terms of DeSync and FAD at various sampling steps (including 8) against MMAudio, we have included a more extensive comparison against Frieren (reflowed) at 4 sampling steps. We observe that SALSA-V maintains its performance in the few-step sampling setting.
>
> |         | Steps | FAD_VGG  | KL_PANNs | KL_PaSST | IS        | IB        | DeSync    |
> |---------|-------|----------|----------|----------|-----------|-----------|-----------|
> | Frieren |     4 |     2.17 |     2.54 |     2.52 |      9.46 |      22.1 |     0.917 |
> | SALSA-V |     4 | **1.19** | **1.83** | **1.67** | **14.57** | **31.18** | **0.536** |
>
> > The evaluation benchmark is vaguely described. For reproducibility, could the authors provide a precise composition and list of identifiers for their custom test set?
>
> We used multiple sources of data to evaluate our model, both to better reflect the variety of in-the-wild videos (beyond those classes included in VGGSound) and because the test sets commonly used in V2A are of a short fixed length (e.g. 10 seconds for VGGSound). This includes videos from the test sets of VGGSound, UnAV-100, and additional videos from Panda-70M. Due to licensing issues, we can only publish the video identifiers and timestamps and not the videos themselves. This list of evaluation IDs and timestamps will be released alongside the codebase as part of the model release.

---

### Official Review · Reviewer_7eea · 2025-11-01

**Soundness:** 3
**Presentation:** 2
**Contribution:** 3
**Rating:** 6
**Confidence:** 4

**Summary:**

This paper introduces SALSA-V, a novel video-to-audio generation model that synthesizes high-fidelity, temporally aligned long-form audio from silent videos. The model leverages a masked diffusion objective to support audio-conditioned generation and seamless outpainting, enabling the synthesis of audio sequences of arbitrary length. A key innovation is the integration of a shortcut loss during training, which allows for high-quality audio generation in as few as eight sampling steps without additional fine-tuning. Furthermore, the authors introduce a contrastively-trained synchronization module using a large-scale pretrained vision backbone, which significantly improves temporal alignment between visual events and generated sounds. Extensive evaluations demonstrate that SALSA-V outperforms existing state-of-the-art models in both objective metrics (e.g., DeSync, FAD) and human subjective ratings, particularly in synchronization and long-form generation quality. The model also maintains competitive performance in semantic alignment and audio fidelity, despite having fewer parameters than leading baselines. These contributions collectively address key limitations in current V2A systems, including generation speed, controllability, and scalability to long durations.

**Strengths:**

* High-Quality Synchronization: The model demonstrates state-of-the-art performance in temporal alignment between audio and video, as evidenced by both objective metrics (DeSync) and human evaluation. This is a critical and challenging aspect of video-to-audio generation.
* Efficient Few-Step Sampling: By incorporating a shortcut loss during training, the model achieves high-quality audio generation in as few as eight sampling steps, without requiring additional fine-tuning or distillation. This makes it suitable for near-real-time applications.
* Long-Form Generation Support: Through a masked diffusion objective, the model supports audio conditioning and seamless outpainting, enabling the generation of synchronized audio for extended video sequences without significant performance degradation.
* Strong Multimodal Conditioning: The model effectively leverages multiple conditioning sources—semantic visual features, high-resolution synchronization features, and text embeddings—using a modified multimodal transformer architecture, leading to improved semantic and temporal alignment.

**Weaknesses:**

* Evaluation of Contrastive Pre-training Models: The CAVP model from Diff-Foley is also a pre-trained model for audio-visual alignment, yet the authors omitted a comparative analysis with it. Could an experiment be designed to compare the performance of these two models? Additionally, while the paper discusses the impact of different batch sizes on contrastive learning, it lacks an ablation study to substantiate these claims.
* Masked Training Strategy: During inference, only the preceding segment of clean audio is prepended. Why, then, is the masking during training applied randomly rather than being restricted to the preceding portion as well? Is there any prior investigation or ablation experiment regarding this design choice?

**Questions:**

Please refer to the weaknesses section.

**Details Of Ethics Concerns:**

No concerns.

---

> ### Author Response · Authors · 2025-11-20
>
> We thank the reviewer for their feedback. In the following we address each concern.
>
> > The CAVP model from Diff-Foley is also a pre-trained model for audio-visual alignment, yet the authors omitted a comparative analysis with it. Could an experiment be designed to compare the performance of these two models? Additionally, while the paper discusses the impact of different batch sizes on contrastive learning, it lacks an ablation study to substantiate these claims.
>
> Since our work focuses on improving audio-visual synchronization, our contrastive model was specifically trained for this task, unlike CAVP, which includes a semantic alignment component.
> Regarding the choice of batch size, a similar finding was made in Synchformer [1], which includes a small investigation into increasing the training batch size, which resulted in decreased synchronization performance.
>
> > During inference, only the preceding segment of clean audio is prepended. Why, then, is the masking during training applied randomly rather than being restricted to the preceding portion as well? Is there any prior investigation or ablation experiment regarding this design choice?
>
> Randomly masking within the entire sequence enables both audio conditioning and extension during inference. Only masking the beginnings of sequences would retain the extension ability, but would not enable inpainting or more general uses for audio conditioning.
>
> [1] Iashin V, Xie W, Rahtu E, et al. Synchformer: Efficient synchronization from sparse cues[C]//ICASSP 2024-2024 IEEE International Conference on Acoustics, Speech and Signal Processing (ICASSP). IEEE, 2024: 5325-5329.

---

### Author Response · Authors · 2025-12-02

We thank all reviewers for the time spent reviewing our work and for their thoughtful comments. In the following we summarize the discussion and the additional results.

SALSA-V achieves strong temporal synchronization and high-quality audio generation at few sampling steps, while displaying superior long-form generation capabilities against existing methods. Our proposed model also offers audio-conditioned generation, which opens up additional possibilities for video-to-audio generation workflows.

Thanks to reviewer feedback we have improved readability of various sections in the manuscript and added additional baseline comparisons against Frieren, V-AURA, LoVA, and AudioX. The results show that SALSA-V maintains its state-of-the-art performance against these new baselines.

In addition, we conducted a long-form generation comparison against LoVA, as well as a few-step comparison against Frieren at 4 sampling steps, showing superior performance in both settings.

Furthermore, we included an additional runtime comparison against multiple models, showing that SALSA-V displays a superior quality-efficiency tradeoff compared to existing baselines.

We believe that by including these additional results, we have strengthened our key claims of SALSA-V achieving superior synchronization and quality compared to the current state-of-the-art model, especially in the long-form and few-step regime.

---

### Meta-Review · Area_Chair_mFbZ · 2026-01-09

**Summary:**

Reviewers generally acknowledge that the paper presents a well-engineeredvideo-to-audio system, with strong empirical performance in audiovisualsynchronization, long-form generation, and efficient few-step sampling.The main concern raised by multiple reviewers relates to 1imited technical novelty.Reviewer jLCz notes that "the paper, while presenting a well-engineered system,suffers from several weaknesses that question the significance and novelty of its
contribution".Several reviewers pointed out that the experimental comparisons were insufficient.Reviewer jLCz noted that "the paper's experimental comparison is narrow," whileReviewer gTEV commented that "the comparison in the experiments needs additionalbaselines.'
Similarly, Reviewer gTEV comments that "the novelty of the methodology ismarginal".Concerns were also raised regarding evaluation clarity. Reviewer jLCz pointsout a "lack of clarity in experimental details", while Reviewer 7VCU notes a "mismatchbetween the performance trend in Figure 5 and the metrics in Table 1".

**Reviewer Concerns:**

The authors have added a number of experiments, which makes the experimentalcomparisons much more comprehensive. They also improved the clarity of theevaluation.However, the concerns regarding novelty have not been fully addressed. Inaddition, two issues raised by Reviewer gTEV remain unresolved: (1) whether theexample shown in Fig. 4 is appropriate for demonstrating the advantages of theproposed method, and (2) whether there is any reason why the authors did not try1B-parameters.

**Reviewer Scores:**

Reviewer 7eea was relatively positive in the initial review, so this reviewerwould likely maintain the original score.Reviewer jLCz raised concerns primarily related to experimental coverage andclarity, many of which were at least partially addressed in the rebuttal. This couldplausibly lead to a modest upward adjustment of the score.Reviewer gTEV and Reviewer 7VCU participated in each round of the discussion andhad already acknowledged the rebuttal and follow-up clarifications. Therefore, theirscores are unlikely to change further.

---

### Decision · Program_Chairs · 2026-01-26

Reject